# GAVEL: Generating Games Via Evolution and Language Models

**Graham Todd**
New York University Tandon
Brooklyn, New York, USA
gdrtodd◇

**Alexander G. Padula**
ETH Zurich
Zurich, Switzerland
apadula♠

**Matthew Stephenson**
Flinders University
Adelaide, Australia
matthew.stephenson♣

**Éric Piette**
UCLouvain
Louvain-la-Neuve, Belgium
eric.piette♡

**Dennis J.N.J. Soemers**
Maastricht University
Maastricht, the Netherlands
dennis.soemers†

**Julian Togelius**
New York University Tandon
Brooklyn, New York, USA
julian.togelius◇

◇@nyu.edu, ♠@ethz.ch,
♣@flinders.edu.au, ♡@uclouvain.be, †@maastrichtuniversity.nl,

## Abstract

Automatically generating novel and interesting games is a complex task. Challenges include representing game rules in a computationally workable form, searching through the large space of potential games under most such representations, and accurately evaluating the originality and quality of previously unseen games. Prior work in automated game generation has largely focused on relatively restricted rule representations and relied on domain-specific heuristics. In this work, we explore the generation of novel games in the comparatively expansive `Ludii` game description language, which encodes the rules of over 1000 board games in a variety of styles and modes of play. We draw inspiration from recent advances in large language models and evolutionary computation in order to train a model that intelligently mutates and recombines games and mechanics expressed as code. We demonstrate both quantitatively and qualitatively that our approach is capable of generating new and interesting games, including in regions of the potential rules space not covered by existing games in the `Ludii` dataset. A sample of the generated games are available to play online through the `Ludii` portal. [1]

## 1   Introduction

Games have long been used as a test bed for algorithms and approaches in artificial intelligence, with advances in game-playing ability often serving as some of the most recognizable achievements in the field [13, 40, 55, 65]. While automated systems have repeatedly demonstrated an ability to match or surpass humans as game players, they continue to lag significantly in their capacity to generate the kinds of games that are worth playing. The ability to construct novel and interesting games is an impressive cognitive challenge, and success would have implications both cultural (with the

---

[1]Play the generated games at: https://ludii.games/details.php?keyword=Havabu, https://ludii.games/details.php?keyword=HopThrough, and https://ludii.games/details.php?keyword=YavaGo

38th Conference on Neural Information Processing Systems (NeurIPS 2024).

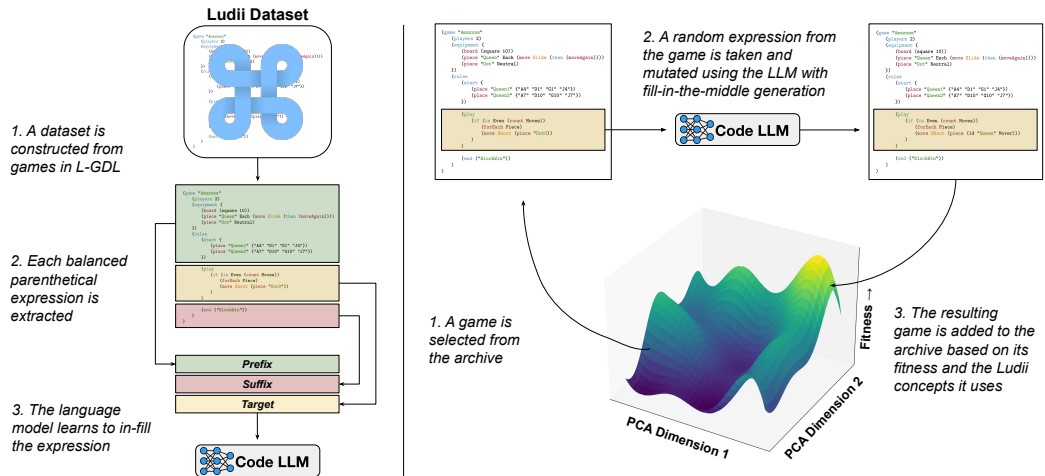

Figure 1: **GAVEL overview. Left:** a dataset of games in the `Ludii` game description language is used to train a code large language model using the *fill-in-the-middle* objective on parenthetical expressions. **Right:** the trained code language model can then be used as the mutation operator for evolutionary quality-diversity optimization with the MAP-Elites algorithm. Fitness is determined with a suite of automatic evaluation metrics, and the `Ludii` game description language also affords a large number of semantic game "concepts" that are used to determine game novelty.

production of new artifacts) and computational (with the generation of new learning environments for artificial agents). Prior efforts in *automated game design* [42, 62] have produced some successes (notably the commercially-available game *Yavalath*, generated by the `Ludi` system [12]), but remain largely limited by hard-coded heuristics and restricted domains. These limitations are often necessary, however, in the face of automated game design's core challenges: (1) representing the vast array of possible games in a structured and computationally workable form and (2) efficiently searching through the resulting representation space for worthwhile games.

In this work, we present GAVEL (**Ga**mes **v**ia **E**volution and **L**anguage Models)—an automated game design system that tackles these challenges by leveraging recent improvements in game rule representation and code synthesis (see overview in Figure 1). GAVEL draws on three main components: (1) the `Ludii` game description language [11, 46] to efficiently encode a large variety of board game rule sets, (2) a large code language model to reliably produce plausible modifications to existing games inspired by *evolution through large models* (ELM) [34], and (3) *quality-diversity* optimization [49] to generate a wide range of playable and interesting games. Each component builds on the others: our choice of representation allows GAVEL to not only produce novel board games in a wide range of genres and styles, but also affords us a dataset of over 1000 existing board games from around the world [10]. This dataset, in turn, provides sufficient basis to *fine-tune* a code synthesis model. In addition, our quality-diversity approach leverages the inherently modular nature of our representation in order to determine game novelty through the presence of particular game mechanics and motifs.

We show empirically that GAVEL is capable of generating playable and interesting board games that differ substantially from games encountered during training. Our approach intelligently recombines mechanics and ideas from disparate genres and produces samples that mirror the performance of human-generated games under a suite of automated evaluation metrics. A preliminary qualitative analysis also reveals that GAVEL can generate novel games that are both engaging and entertaining. We conclude with a discussion of GAVEL's successes and failures, as well as the promising avenues for future work. We provide a public repository that includes our code and data, including a trained model checkpoint. [2]

---

[2]Code and data available here: `https://github.com/gdrtodd/gavel`

## 2 Related Work

### 2.1 Automatic Game Generation

Our work continues a strand of research that investigates the ability for automated systems to produce novel games or novel variants of existing games. The first such effort was METAGAME [45], which samples from a grammar that encodes "symmetric *Chess*-like games" with designer-specified rule probabilities. Since then, work has continued in the generation of both board game and video game rulesets. Evolutionary or search-based game design is a popular technique, as game descriptions do not typically afford gradient information; it was first proposed in 2008 for board games [12] and video games [62], and later work has brought it to bear on different video game genres [43, 16, 29]. Another approach begins instead from conceptual or symbolic specifications of rules or mechanics and generates games by dynamically referring to a pre-specified library of gameplay elements [42, 63]. Yet another approach is to use constraint satisfaction algorithms. For instance, rules might be encoded as an answer-set program, with constraints pre-specified by a designer to define what counts as an acceptable game [56, 71, 44, 26, 59]. Most recently, work has investigated the ability for large language models to act as design assistants by generating game levels [61, 58], proposing game mechanics [3] or directly synthesizing small programs [27].

### 2.2 Evolutionary Computation and Language Models

Evolutionary computation refers to a large class of algorithms broadly inspired by the biological process of evolution [24]. Of these, our approach descends most directly from genetic programming: the use of evolution or other stochastic search procedures for program synthesis [18, 23, 32]. We also draw on more recent advances in quality-diversity algorithms that aim to find a distribution of solutions to a given problem rather than a single optima [41, 49, 20], especially in the context of code and content generation [25].

In recent years, improvements in large language models both generally [7, 70] and in their ability to produce code [14, 67] have led to their use in a variety of evolutionary systems. Of note is *evolution through large models* [34], which learns a *diff model* (i.e. a language model that can modify programs conditioned on a natural language specification) from a dataset of GitHub commits and accompanying messages. This model is then used as the mutation operator for genetic programming in Python through a quality-diversity algorithm [41]. We adopt this general approach, though GAVEL learns to mutate games from raw programs (i.e. without diffs or commit messages) in a domain-specific language. Similar techniques have also been used to generate reinforcement learning environments [1, 69] and reward functions [38], adversarial prompts [53], programming puzzles [48], and poetry [6].

## 3 Game Representation and Dataset

We represent games as programs in the Ludii game description language (L-GDL) [11], in which game rules are built from *ludemes*—high-level keywords that represent common components in the natural language descriptions of board game rules. Examples of such keywords include step, slide, hop, piece, empty, board, and so on. Owing to this abstraction, the L-GDL is both robust enough to encode a vast array of disparate games [46] and compact enough that game descriptions often fit within the context lengths of modern large language models.

In addition to *ludemes*, the Ludii system also defines a large number of *concepts*—high level properties of games that describe its gameplay or structure [47]. Most concepts are boolean and indicate the presence or absence of a particular feature. For example, one concept might describe whether a game is asymmetric, while another might describe whether a game uses the "custodial" capture mechanic seen in *Tafl*-style games. These concepts provide a way to represent games as meaningful feature vectors, which can then be used to cluster and compute similarities between games [57].

### 3.1 Dataset

We construct our initial game dataset out of the 1182 existing games that have been translated into the Ludii game description language (available under a Creative Commons BY-NC-ND 4.0 license).

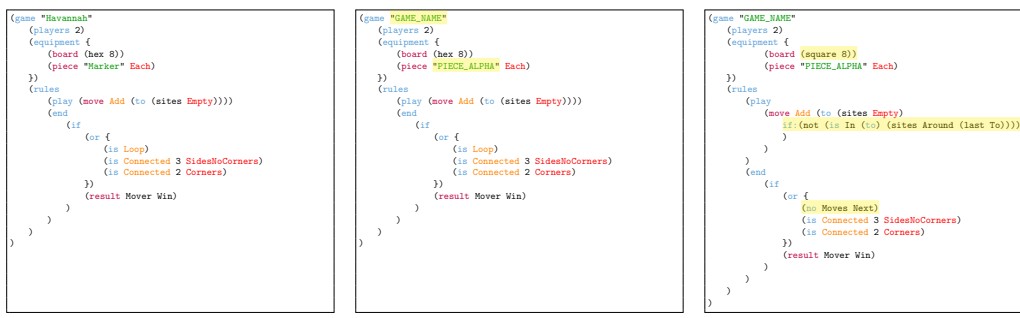

Figure 2: **Left:** the game of *Havannah* by Christian Freeling rendered in the `Ludii` game description language. **Center:** the same game as it appears in the training dataset, with functional references expanded and game / piece names replaced with abstract identifiers. **Right:** a variant of *Havannah* produced by `GAVEL`. Changes are highlighted in yellow.

We start by expanding the function references (e.g. `"BlockWin"`) inside each game description to the code they represent (e.g. `(end (if (no Move Next) (result Mover Win)))`), as specific functions may appear in only few games while the underlying *ludemes* are much more widespread. In addition, we remove references to the particulars of each game (i.e. its name and the names of the pieces it uses) and replace them with abstract identifiers. Both these processes increase the generality of our game description dataset.

After this, we filter our dataset to remove a small number of puzzles and experimental games. We also remove *Mancala*-style games, as prior work has identified them as occupying a very distinct cluster with respect to the full collection of `Ludii` games, in terms of rules and structure [57]. We then tokenize each game according to our code language model (see below) and exclude any game that is longer than 1024 tokens. From this reduced dataset, we hold out a set of 14 varied games (available in Appendix A) that are used to initialize the evolutionary search, with the remaining 574 games being used as our training dataset. An example of one of the held-out games and its converted form is available in Figure 2.

## 4 Methods

### 4.1 Language Model Training

In line with the general ELM approach, we make use of the impressive generative capabilities of modern code language models in order to propose sensible modifications to existing programs [34]. Unlike prior work, however, we specifically fine-tune an existing model to operate in L-GDL instead of working with programming languages seen during pre-training or making use of in-context learning. In addition, we train our model to act as a mutation operator over programs by using a *fill-in-the-middle* (FITM) [5] training objective instead of the more common left-to-right objective. FITM training allows the model to make changes to interior components of a program without (a) relying on an extant dataset of code diffs or (b) regenerating the entire game at each step.

We train an instance of `CodeLlama` [52] (specifically `CodeLlama-13b`, as it is the largest model in its family that was pre-trained with a FITM objective) on the dataset described in Section 3.1. To facilitate FITM training, we extract every balanced parenthetical expression from each game (e.g. `(board (square 10)))`) and add it to the dataset along with the corresponding prefix and suffix in the program. With respect to the grammar of L-GDL, this process is equivalent to extracting syntactic nodes and all of their descendants. The final dataset consists of 49,968 such (prefix, suffix, target) tuples. To facilitate training on a single GPU, we make use of both parameter-efficient fine-tuning [39] and 8-bit quantization [22]. Owing to the large size of the dataset and the fact that each game appears repeatedly in different configurations, we fine-tune the model for a single epoch with hyperparameters available in Appendix B. Training took approximately 40 hours to complete on a single RTX8000 GPU.

We note here that FITM training may have a potential downside in the context of evolution through large models. Specifically, the model is trained to perfectly reproduce the missing section of code given its prefix and suffix. If it succeeds completely in doing so during evolution, then the resulting game will not be mutated at all. In essence, there is a tension between the ability of the model to accurately capture the underlying logic and syntax of the representation space and its ability to memorize or perfectly reconstruct its training dataset. In `GAVEL`, we mostly sidestep this issue by mutating a set of held-out games not seen at all during training (making memorization impossible), though see Appendix C for an initial investigation on mutating training-set games and Section 8 for a discussion of other possible approaches.

## 4.2 Evolutionary Search

Our evolutionary search strategy of choice is MAP-Elites [41], a population-based quality-diversity algorithm that leverages both a fitness function over samples as well as a set of *behavioral characteristics*—functions that describe non-fitness attributes of samples and are used to ensure that the population does not collapse to only a small number of distinct samples. Specifically, MAP-Elites maintains an *archive* of cells, each associated with a particular range of values under the behavioral characteristics. At each step, a novel sample is evaluated to determine its fitness as well as the cell it would occupy. It is added to the archive if either that cell is unoccupied or if its fitness exceeds that of the current occupant, in which case it replaces the current occupant. In this way, samples only "compete" with one another within particular cells. We describe our fitness function in detail in Section 4.3.

Determining an appropriate set of behavioral characteristics is challenging in the context of automatic game generation. Intuitively, distinct archive cells ought to capture meaningfully distinct games while collapsing minor or trivial variations. Human game players readily make such categorizations, but automatically identifying such differences (especially with previously unseen games) is both difficult and necessary for the MAP-Elites algorithm to function. In order to tackle this problem, we take advantage of the semantic concepts described in Section 3. Building on evidence that `Ludii` concept vectors capture a meaningful notion of distance [57], we use principal component analysis (PCA) [66] on the complete `Ludii` dataset (i.e. before any filtering) to reduce the 510-dimensional concept vectors to two dimensions. We then bucket the resulting two-dimensional space into 40 equally-spaced regions from -5 to 5 in each dimension, obtaining a rectangular archive of 1600 cells. While the first two PCA dimensions describe only $\sim 28\%$ of the variance in the concept feature space, a preliminary investigation indicated that increasing the number of dimensions resulted in a *less* diverse archive (see Appendix D for additional details). Because games are not uniformly distributed through feature space, increasing the archive's dimensionality for a fixed number of cells caused a larger number of distinct games to be mapped to the same cell. See discussion of potential alternatives in Section 7.

We initialize the archive by adding and evaluating the 14 heldout games listed in Appendix A. For each MAP-Elites step, we select $j$ games from the current archive. For each game, we then select $k$ random parenthetical expressions and re-format them as a (prefix, suffix, target) tuple. We then sample from the trained `CodeLlama-13b` model with a temperature of 1 and a top-$k$ value of 50 to generate a new expression, conditioned on just the prefix and suffix, and re-construct the resulting game. After filtering out any duplicate or unchanged games, the samples are evaluated for fitness and assigned an archive cell based on the PCA reduction of their concept vector.

## 4.3 Evaluation

Game quality is both difficult to quantify and inherently subjective. Nevertheless, automatic game design and evolutionary computation necessitate some kind of computable optimization objective. These objectives typically take the form of one or more heuristics that aim to proxy the underlying targets of "fun" or "interestingness." For restricted domains, it is often possible to imbue a large amount of expert knowledge into these heuristics, as in the `Ludi` system [12] (precursor to `Ludii`) which employed game-state evaluator functions to capture both objective measures (e.g. a game's balance) and psychological measures (e.g. a game's excitement or unpredictability). However, the large space of games described by the `Ludii` description language makes relying on such evaluator functions infeasible. Instead, we define a hierarchical fitness function based on a relatively small set of objectively and reliably measurable heuristics that are fully game-agnostic.

**Algorithm 1** `GAVEL` Game Evaluation

---

**Input:** a game $g$ in L-GDL
**Output:** a fitness evaluation $f(g) \in \{-3, -2, -1\} \cup [0.01, 1]$

  **if** $\neg$`compilable`$(g)$ **then**
     **return** -3
  **else if** $\neg$`playable`$(g)$ **then**
     **return** -2
  **else**
     $e \leftarrow$ `random_eval`$(g, n = 100)$
     **if** $e_{\text{balance}} < 0.5$ **or** $e_{\text{agency}} < 0.5$ **then**
       **return** -1
     **else**
       $v \leftarrow$ `mcts_eval`$(g, n = 10, t = 0.25, m = 50)$
       $d \leftarrow$ `strategic_depth`$(g, n = 10, t = 0.25, m = 50)$
       $f \leftarrow$ `hmean`$(v_{\text{balance}}, v_{\text{decisiveness}}, v_{\text{completion}}, v_{\text{agency}}, v_{\text{coverage}}, d)$
       **return** $f$
     **end if**
  **end if**

---

Concretely, every game $g$ generated during the evolutionary search is assigned a fitness value $f(g) \in \{-3, -2, -1\} \cup [0.01, 1]$ by a sequence of tests (pseudo-code presented in Algorithm 1). Evaluation begins with a series of binary evaluations. First, all games that fail to compile (e.g. due to grammatical errors, or because they do not define a board) are assigned the minimum possible fitness of -3 (we note that such games cannot be added to the archive as their concepts are not well defined). Next, all compilable games that cannot be played (e.g. due to failing to define any moves for pieces or failing to place pieces on the board for a game like *Chess*) are assigned a fitness of -2. If these conditions are cleared, then random-policy agents are used to rapidly obtain $n = 100$ playouts of the game. If the difference in win rate between the first and second player is larger than a threshold of 0.5 (indicating a large imbalance even for completely naive players) or if fewer than half of game states allow more than one legal move (indicating a lack of player agency), then the game is assigned a fitness of $-1$. These checks are used as a filter to ensure that expensive evaluations are only performed on promising games.

If all three previous conditions are met, then search-based agents are used to obtain $n = 10$ game playouts. Specifically, we use self-play between Monte-Carlo Tree Search (MCTS) [31, 17, 9] agents with $t = 0.25$ seconds of thinking time per move and a hard limit of $m = 50$ moves per player—any game which exceeds the limit is terminated and called a draw. This limit puts a strong pressure on games to end within a reasonable number of moves and necessarily excludes many potentially interesting games (especially as search-limited agents might fail to find winning lines that could end a game quickly). However, this concession is necessary to ensure that the overall evaluation procedure remains computationally tractable.

From these 10 playouts, we extract the following evaluation metrics, largely inspired by prior work in automated game design [2, 51, 12]:

1. **Balance**: the largest difference in winrates between any pair of players.
2. **Decisiveness**: The proportion of games that do not end in a draw.
3. **Completion**: the proportion of games that reach an end state.
4. **Agency**: the proportion of turns for which the player to move has more than one legal move.
5. **Coverage**: the proportion of board sites (e.g. squares on a chessboard) that get occupied by a game piece at least once in a playout.

In addition, we separately compute a final metric: **Strategic Depth**, defined as the proportion of games won by an MCTS agent against a random agent over $n = 10$ playouts (and inspired by previous similar evaluations [43, 8, 33]). Each evaluation metric returns a value between 0 and 1, and the metrics are then aggregated into a single fitness score by taking the *harmonic mean*. We use the harmonic mean over the simple average because the former is weighted towards small values, penalizing games that succeed in most metrics but fail dramatically in one. We enforce a minimum

| Method | QD Score | All Cells | | Novel Cells | |
|---|---|---|---|---|---|
| | | # Playable | # Fitness>0.5 | # Playable | # Fitness>0.5 |
| GAVEL | 395.62 ± 17.46 | 117.67 ± 9.46 | 106.67 ± 7.41 | 26.67 ± 3.30 | 21.67 ± 6.13 |
| GAVEL-UCB | 341.17 ± 14.39 | 96.67 ± 6.02 | 88.33 ± 7.41 | 19.67 ± 4.03 | 16.00 ± 2.45 |
| Pure Sampling | 296.92 ± 14.84 | 89.00 ± 5.66 | 83.00 ± 5.10 | 14.67 ± 3.30 | 11.33 ± 2.87 |
| GPT-4o | 268.16 ± 17.33 | 84.67 ± 6.60 | 80.67 ± 5.19 | 16.67 ± 3.30 | 15.33 ± 2.87 |

Table 1: Quantitative measures of archive progress for GAVEL, a variant in which mutation locations are selected with the UCB algorithm, and two baseline methods, averaged over three independent runs. We report the quality diversity (QD) score (a cumulative measure of fitness) as well as the number of archive cells and novel archive cells that reach certain fitness thresholds. Both GAVEL-based methods succeed in producing high-fitness games in unexplored regions of concept space, though GAVEL has the edge over GAVEL-UCB in overall QD score. Compared to baseline methods, GAVEL achieves a significantly higher QD score and fills more of the archive will playable and high-fitness games.

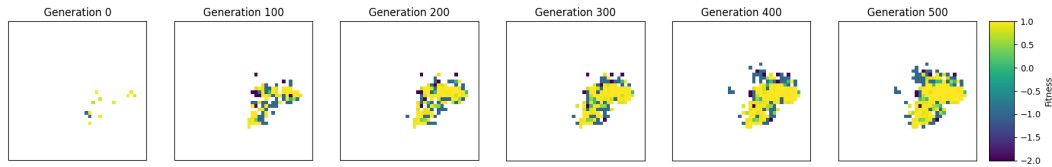

Figure 3: A visualization of the fitness of games generated by GAVEL over time. Starting from an initial archive of 14 games, GAVEL produced in this run 185 novel variations within 500 generations, of which 130 are playable and meet our minimum evaluation criteria. Further, 62 generated games occupy cells not covered by *any* game in the Ludii dataset and 29 of these games meet our minimal criteria.

metric value of 0.01 before taking the harmonic mean to ensure that the fitness score is not completely zeroed-out by a single metric.

## 5 Experiments

We perform 3 runs of MAP-Elites with random seeds $\{1, 2, 3\}$, each lasting for 500 steps. For each run, we select $j = 3$ games and generate $k = 3$ mutations for each game at each step. Quantitatively, we report the progress of the archive using three metrics: the *quality-diversity score* (QD score) [50], calculated as the sum of the fitness of each cell in the archive. In cases like ours where fitness values can be negative, each fitness value is incremented by the minimum possible fitness (i.e. -2) before being summed to ensure that the QD score increases monotonically over time. In addition, we report the number of playable and minimally interesting games (i.e. $f(g) > 0$) in the archive as well as the number of such games that occupy cells which are not covered by any game in the Ludii dataset. Finally, we report the number of cells and novel cells that contain games with a fitness of at least 0.5, indicating their potential as worthwhile games. Each run lasted roughly 48 hours using a single RTX8000 GPU for inference from the CodeLlama-13b model and performing evaluations in parallel with 16 CPU cores and 128GB of total memory.

In addition, we perform another set of 3 runs with a variant of GAVEL that uses the Upper Confidence Bound algorithm [4] to select which regions of games to mutate. Specifically, we treat the selection of a parenthetical expression to mutate as a multi-armed bandit problem where each arm corresponds to a different leading *ludeme* (e.g. board or equipment). We consider a mutation "successful" if it results in the mutated game being added to the archive (either by improving the fitness of an existing occupant, or by occupying a new cell) and update the statistics for each "arm." We call this variant GAVEL-UCB. All other hyperparameters remain the same as with the original GAVEL experiment.

We compare GAVEL against two baselines: a pure sampling approach, which uses the same fine-tuned large language model but omits the quality diversity search, and direct sampling from GPT-4o. For the pure sampling baseline, we randomly select a game from the validation set and mutate it by re-generating a random parenthetical expression. We repeat this process 4500 times in order to match the number of samples produced by GAVEL in 500 generations with $j = 3$ and $k = 3$. We then

evaluate the set of 4500 samples for fitness and determine the cell that each game would occupy based on its concepts, allowing us to construct a simulated archive (i.e. by retaining the highest-fitness sample in each cell) and compare directly against GAVEL. We perform the pure-sampling baseline experiment three times with random seeds $\{1, 2, 3\}$. For the GPT-4o baseline, we provide the model with each of the 14 validation games as part of the context window and then ask it to create a modification of one randomly-selected validation game (see Appendix E for prompt). We similarly generate 4500 samples and construct a simulated archive in order to facilitate comparisons with GAVEL.

# 6   Results

## 6.1   Quantitative Results

Overall, GAVEL succeeds in generating a wide range of novel and high-fitness games. In Table 1 we present the mean and standard deviations of the archive metrics for GAVEL, GAVEL-UCB, and our baseline methods. The quality diversity score is difficult to interpret in isolation, as its magnitude depends greatly on the potential range of fitness scores. In this case, it is most helpful as a way to compare the performance of disparate algorithms on the same task: we see that GAVEL improves significantly over GAVEL-UCB (Welch's $t$-test, $p = 0.029$), indicating that it has some mixture of higher fitness and greater variety in the samples it produces. Similarly, GAVEL improves significantly in terms of QD score over both the pure sampling baseline ($p = 0.004$) and the GPT-4o baseline ($p = 0.002$).

Of more interest is the fact that GAVEL fills a substantial proportion of the archive with playable games despite starting from a modest 14 samples, including in regions of the archive not covered by games in the Ludii training dataset. Compared to both baselines, GAVEL occupies significantly more cells in the archive and produces more high-fitness (i.e. $f(g) > 0.5$) samples ($p < 0.03$ for pure sampling, $p < 0.02$ for GPT-4o), though the differences between GAVEL and GAVEL-UCB are not significant ($p > 0.05$). While GAVEL also appears to occupy a larger number of *novel* cells with playable and high-fitness games compared to the baselines, these improvements are not statistically significant at only three runs. We present a visualization of the archive produced by one run of GAVEL over time in Figure 3, which shows both the success of the model in generating high-fitness samples and the fact that much of the concept space remains unexplored.

## 6.2   Qualitative Results

In order to more closely examine GAVEL's output, we rely on expert evaluators to quickly playtest potentially promising games. These evaluators are broadly familiar with the Ludii dataset and so are able to determine whether a generated game is truly novel and where its mechanics might have originated from. This preliminary human analysis helped shed light on some of GAVEL's shortcomings (discussed below) and also revealed some particularly interesting games among the high-fitness samples. Of special note is a variant of *Yavalath*, itself the product of an automated system [12]. In the original game, players take turns placing pieces on a hexagonal board. A player wins if they have four pieces in a row but loses if they have three pieces in a row first. GAVEL makes both minor changes to the ending rules (increasing the number of pieces in a row needed for victory and loss by one) as well a substantial addition by introducing the enclosure capture rules of *Go*. The result is a game that tasks players with thinking about the arrangement of their pieces in many ways and that appears to offer the potential for sophisticated strategy. In Figure 4 we present an example of play between automated agents in which all of the game's rules are used in concert.

Figure 2 (right) includes another example of a generated game noted by our evaluators to be particularly interesting. It is a variant of *Havannah* (a game in which players attempt to form loops or connected lines of pieces between sides of the board) that introduces a restriction on piece placement from another game in the Ludii dataset (*Tabu Y*). A final exemplar, presented in Appendix F alongside the previous two examples, modifies the pawn-advancement game *Breakthrough* to use pieces that can only move by hopping over each other (and without capturing). Taken together, these examples demonstrate the strength of GAVEL: it is able to intelligently recombine game mechanics (expressed as code segments) in novel ways and ensure that these combinations do not result in trivial games.

```
(game "YavaGo"
    (players 2)

    (equipment {
        (board (rotate 90 (hex 5)))
        (piece "Marker" Each)
    })

    (rules
        (meta
            (no Repeat)
        )
        (play
            (move Add
                (to (sites Empty))
                (then
                    (enclose
                        (from (last To)) Orthogonal
                        (between if:(is Enemy (who at:(between)))
                            (apply (remove (between)))
                        )
                    )
                )
            )
        )

        (end {
            (if (is Line 5) (result Next Loss) )
            (if (is Line 4) (result Next Win))
        })
    )
)
```

*Players take turns placing markers on a hexagonal board. Getting 5 pieces in a row **wins**, but getting 4 in a row **loses**. Surrounding your opponent's pieces removes them.*

***1.*** *What seems to be a relatively balanced position…*

***2.*** *Black moves, capturing many pieces and setting up a lethal threat…*

***3.*** *White makes the only possible play to not lose on the spot…*

***4.*** *But in a single move, black removes white's defense and sets up a second threat for the win…*

Figure 4: Example of play between MCTS agents in a game generated by GAVEL. The game is descended from *Yavalath* (an $n$-in-a-row style game) and combines a modification of its ending rules with the enclosure capture mechanics of *Go*. Search-based agents reach interesting and strategically deep game positions, hinting at its potential interest to human players as well.

## 7 Discussion and Limitations

**Unused Game Components:** a common failure mode is that the model will generate game rules that are not actually used during gameplay. For instance, a change to the equipment section might add dice as additional game pieces. However, without further changes to the gameplay section to incorprate them, the dice will remain unused. Detecting such extraneous rules automatically is challenging, as the Ludii system does not provide a way to determine which rules are activated during a given playout. In addition, penalizing games with unused components during fitness evaluation might *harm* diversity by eliminating potential "stepping stones" to more interesting games. Nevertheless, if changes to the underlying representation *did* allow unused sections to be detected, it might be possible to bias future mutations towards relevant gameplay sections in order to increase the likelihood of the missing rules being generated.

**Heldout Games:** as noted in Section 4.2, we initialize the search using a set of games held out from the language model training in order to prevent memorization. While GAVEL produces a wide range of games from this set, they nonetheless will share many features with one another as a result of their common origins. One potential solution to further increase archive diversity is to improve the variety of mutations, either by increasing sampling temperature or by enforcing novelty with respect to the tokens in the original game section through masking. These techniques might also make it possible to initialize the archive with games in the training dataset, further increasing potential output diversity.

**Archive Selection:** determining an appropriate way to distinguish between games is a general challenge. GAVEL makes use of Ludii concepts, but not all representation schemes afford such detailed semantic information. One promising alternative for such domains is the automatic selection of behavioral characteristics, either through distillation of trajectories obtained during evaluation [19] or by leveraging the ability for large language models to identify archetypes in code [48].

**Evaluation:** our evaluation metrics capture general and *minimal* criteria of interesting games, broadly construed. However, satisfying our evaluation metrics alone is far from *sufficient* evidence for a game being interesting. Ultimately, all metrics in automated game design aim to proxy notions of human preference—as mentioned in Section 6.2, we rely on expert evaluators to filter from high-fitness samples to interesting games. Another worthwhile approach, then, might be to learn these preferences

directly from human ratings [15] or attempt to extract them from latent knowledge in large language models [30].

## 8 Future Work

First and foremost, we are excited about a more in-depth human analysis of the games generated by GAVEL. While our expert evaluators can provide useful insight and analysis, their perspective is ultimately one of many. A larger user study could help identify not only which games are most appealing to human players, but also the particular features of those games that correlate with fun and engagement. This, in turn, could help spur improvements to GAVEL across the board, from archive selection to evaluation metrics.

Within the Ludii domain, one particularly exciting possibility for future work is the integration of the explicit L-GDL grammar. While the CodeLlama-13b model learns to produce syntactically valid mutations, integrating the grammar either through token masking [54] or Monte-Carlo steering [35] could allow for greater mutation diversity (by increasing sampling temperature, for instance) without sacrificing syntactic correctness and compilability. In addition, it might be possible to use data-augmentation techniques (e.g. sampling from the underlying grammar) to create a dataset of Ludii game diffs with which to train a more typical ELM model.

More generally, large language models could also be used to link natural language descriptions of game rules with their programmatic representations. For example, an instruction-tuned model could be used to convert from abstract rules to executable code, either by fine-tuning (e.g. on the Ludii dataset) or through in-context learning. This might allow automatic game design systems to interact with and generate games at the level of natural language, better resembling the process used by human designers, while still retaining the ability to automatically evaluate the relevant gameplay properties of those games.

Finally, our results indicate that systems like GAVEL might be most useful in the context of co-creativity [21, 68]. Automated processes are able to rapidly generate plausible combinations of game mechanics, while human designers and play-testers are able to much better determine the subtle changes necessary to elevate a potentially interesting idea to an entertaining and engaging game. A system which explicitly integrates such expertise may thus prove to be the best way forward.

## 9 Broader Impact

Like all automated game design systems, GAVEL has the potential for impacts on the larger space of game design. Especially at time of writing, as the video game industry experiences widespread layoffs and contractions, it is important that automatic systems are used to assist and inspire human designers instead of replacing them. Indeed, our results indicate that such collaboration is crucial to the generation and identification of worthwhile games. We also draw attention to GAVEL's use of large language models: while the Ludii dataset is publicly available, a similar system could conceivably be used to generate games from scraped datasets without such free access or from game code encountered during pre-training. Special care should also be taken to ensure that language model outputs are manually verified before being published. Overall, however, we feel that the specific domain, use case, and outputs of GAVEL mean that its broader impact is unlikely to be negative.

## Acknowledgments

This material is based upon work supported by the National Science Foundation Graduate Research Fellowship under Grant DGE-2234660. This research is partially funded by the European Research Council as part of the Digital Ludeme Project (ERC Consolidator Grant #771292). The research collaboration was facilitated by COST Action CA22145 - GameTable, supported by COST (European Cooperation in Science and Technology). The authors would like to thank Rushang Gajjal and Sam Earle for their help and constructive feedback throughout the project.

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

# Appendices

## A  Held-Out Games

The following games were removed from the `Ludii` training dataset and used to initialize the MAP-Elites archive in each run, each of which are available in the `Ludii` portal (`https://ludii.games/library.php`):

- *Ard-Ri* (traditional Scottish game)
- *Ataxx* (Dave Crummack & Craig Galley, 1990)
- *Breakthrough* (Dan Troyka, 2000)
- *Gomoku* (traditional Japanese game)
- *Havannah* (Christian Freeling, 1981)
- *Hex* (Piet Hein, 1942)
- *Knightthrough* (probable origin: `http://games.ggp.org/`)
- *Konane* (traditional Hawaiian game)
- *Pretwa* (traditional Indian game)
- *Reversi / Othello* (Lewis Waterman / John W. Mollet, 1883)
- *Shobu* (Manolis Vranas & Jamie Sajdak, 2019)
- *Tablut* (traditional Finnish game [36])
- *Tron* (probable origin: `http://games.ggp.org/`)
- *Yavalath* (`Ludi` system, 2009)

## B  Language Model Training Hyperparameters

The instance of `CodeLlama-13b` used by `GAVEL` was trained with the following hyperparameters:

- **Number of epochs:** 1
- **Batch size:** 1
- **Sequence length:** 1024
- **Optimizer:** `AdamW` [37]
- **Learning rate:** $3 \cdot 10^{-4}$
- **Warmup Ratio:** 0.03

In addition, the model was trained with low-rank adaptation (LoRA) [28] and the following LoRA-specific hyperparameters:

- **`LoRA` Alpha:** 16
- **`LoRA` Dropout:** 0.05
- **`LoRA` r:** 64

## C  Effects of Mutating Training Games

We present a preliminary investigation on the feasibility of initializing `GAVEL` with games from the training dataset rather than the held-out validation set. Using the same fine-tuned `CodeLlama-13b` language model, we generate 300 mutations by sampling a game from the training set, sampling a parenthetical expression from the game, and generating 3 replacements from the model. We perform the same process for the validation set, producing another 300 mutated games. For each mutation, we then compute whether it is an exact duplicate of the originally-sampled game (i.e. whether it is *novel*) and whether the resulting game compiles under the `Ludii` grammar (i.e. whether it is *valid*). We repeat the entire process for different sampling temperatures and settings of $k$ for top-$k$ sampling, the results of which are presented in Table 2.

Unsurprisingly, mutations produced from training-set games are much less likely to be novel (since the model may have memorized the exact target sequence during training). The fact that roughly half of validation-set mutations are also duplicates might seem surprising at first, but it is important to note that many sampled parenthetical expressions are very short (e.g. `(sites)`) and might be the only grammatical possibility at that point the game. Training-set mutations do appear to be compilable more often than validation-set mutations (especially at higher sampling temperatures), but this is likely due at least in part to the increased rates of duplication. The proportion of samples that are both novel *and* valid (i.e. those that represent potentially useful mutations) tells a clear story: sampling mutations from validation-set games is more efficient than doing so from training-set games. A final point of interest is that increasing sampling temperature appears to improve the efficacy of training-set sampling while decreasing the efficacy of validation-set sampling. This further motivates the possible extensions described in Section 7 and Section 8 – high sampling temperature coupled with grammatical sampling constraints might make training-set mutations a viable way forward.

| Temperature | Top-K | Training-set Mutations | | | Validation-set Mutations | | |
|---|---|---|---|---|---|---|---|
| | | *Novel* | *Valid* | *Novel & Valid* | *Novel* | *Valid* | *Novel & Valid* |
| 0.5 | 20 | 17.33 | 99.33 | 16.67 | 47.33 | 98.67 | 46.00 |
| | 50 | 17.33 | 99.33 | 16.67 | 47.33 | 98.67 | 46.00 |
| 1 | 20 | 21.00 | 97.33 | 18.33 | 50.67 | 93.67 | 44.33 |
| | 50 | 21.00 | 97.33 | 18.33 | 49.00 | 94.00 | 43.00 |
| 1.5 | 20 | 33.67 | 88.67 | 22.33 | 56.67 | 76.67 | 33.33 |
| | 50 | 35.66 | 85.33 | 21.00 | 54.33 | 73.67 | 28.33 |

Table 2: The proportion of mutations generated from training-set and validation-set games that are novel, compilable, and both novel *and* compilable. We see that training-set mutations often duplicate the original game, a tendency which is not especially ameliorated by higher sampling temperatures.

## D  Effects of Archive Dimensionality and Size

As part of initial experiments for `GAVEL`, we explored the effect of changing the number of PCA dimensions and the total number of cells used by the MAP-Elites archive on the number of unique cells occupied by training and validation games. In each case, we fit a PCA model with specified number of components on the `Ludii` dataset using the process described in Section 4.2. For a target number of total archive cells $C$ and number of dimensions $D$, we constructed a rectangular archive by separating each axis from -5 to 5 into $\lfloor C^{(1/D)} \rfloor$ evenly-spaced regions. Of course, for many choices of $C$ and $D$, this process produces an archive with a total size much less than $C$ (e.g. for $D = 4$ and $C = 1000$, the archive would have $5^4 = 625$ cells). To combat this, we increased the number of regions in the first dimension as much as possible while keeping the total number of cells no more than $C$ (e.g. for $D = 4$ and $C = 1000$, the number of regions in each dimension is $[8, 5, 5, 5]$ for a total archive size of 1000).

We then simulated adding each of the 574 training games and 14 validation games from our dataset into the archive and measured the number of unique resulting cells. This gives a sense of the "diversity" of the archive – if only a small number of cells are occupied it indicates that the archive fails to

distinguish between substantively distinct games. On the other hand, if the number of occupied cells is roughly equal to the number of games, this might indicate that the archive assigns even almost identical games to different cells and could slow the search process. The results of these experiments are presented in Table 3. Our investigation indicated that increasing the dimensionality for fixed target archive size caused the set of training games to be collapsed to a smaller set of cells – a 2-dimensional archive with 2500 cells is roughly as diverse under this formulation as a 3-dimensional archive with 10000 cells. We ultimately decided to use a 2-dimensional archive with roughly 1500 cells because it was the smallest archive that mapped each of the validation games to distinct cells.

| Input set | Dimension | Target total archive cells | | | | | | |
|---|---|---|---|---|---|---|---|---|
| | | 100 | 500 | 1000 | 1500 | 2500 | 5000 | 10000 |
| Training ($n = 574$) | 2D | 32 | 105 | 166 | 215 | 268 | 370 | 410 |
| | 3D | 24 | 59 | 74 | 101 | 137 | 191 | 247 |
| | 4D | 19 | 52 | 58 | 75 | 109 | 128 | 180 |
| | 5D | 45 | 32 | 48 | 70 | 99 | 98 | 126 |
| Validation ($n = 14$) | 2D | 9 | 13 | 13 | 14 | 14 | 14 | 14 |
| | 3D | 6 | 8 | 9 | 11 | 11 | 10 | 13 |
| | 4D | 5 | 7 | 8 | 10 | 9 | 12 | 12 |
| | 5D | 7 | 7 | 8 | 8 | 9 | 8 | 9 |

Table 3: The number of archive cells occupied by the 574 training set games and 14 validation set games, based on the dimensionality and target total number of cells in the archive. We see that increasing the dimensionality for a fixed target archive size causes the set of training games to be collapsed to a smaller set of cells and that a 2-dimensional archive is the only one that maps each of the 14 validation games to a unique cell.

We also briefly explored the possibility of using a more sophisticated kind of archive (namely a Centroidal Voronoi Tesselation (CVT) based approach [64]) to combat the high dimensionality of the full concept vector space. The CVT archive is designed to scale to high-dimensional problems by breaking down the search space into a pre-specified number of geometrically homogeneous niches, so we attempted to apply it to the full 510-dimensional `Ludii` concept vectors using the implementation in the `Pyribs` library [60] and thereby avoid the need for lossy dimensionality reduction. We set the boundaries in each dimension to $[0, 1]$ because each concept is binary. Unfortunately, even very large archives (i.e. with 100000 cells) collapsed the set of training and validation games into a small number of unique cells. We attribute this to the fact that `Ludii` games are not uniformly distributed throughout the space of concepts (i.e. many concepts are correlated or mutually exclusive) while the CVT algorithm assigns equal "resolution" to all parts of the search space. However, it is possible that some combination of PCA and the CVT archive might achieve a better balance of archive size and diversity.

| Input Set | Number of archive cells | | | | | | | | |
|---|---|---|---|---|---|---|---|---|---|
| | 100 | 500 | 1000 | 1500 | 2500 | 5000 | 10000 | 50000 | 100000 |
| Training ($n = 574$) | 12 | 19 | 34 | 48 | 36 | 37 | 24 | 41 | 68 |
| Validation ($n = 14$) | 3 | 8 | 9 | 7 | 9 | 7 | 4 | 10 | 7 |

Table 4: The number of CVT archive cells occupied by the 574 training set and 14 validation set games, based on the total number of cells in the archive. We see that even very large archives collapse the input sets to a small number of cells.

# E    LLM Baseline Details

As mentioned in Section 5, we use the `GPT-4o` language model through the OpenAI API. To increase the diversity of generated samples, we use a sampling temperature of 1. We specify a unique seed for each sample from the model (i.e. seeds 0 through 4499 for the first experiment), though the OpenAI API does not guarantee reproducibility for a specific seed. We provide the system prompt and user prompt used in the baseline experiments below.

---

**System Prompt**

```
You are an expert programming agent in the Ludii game description
language. You output syntactically correct Ludii game descriptions
 and no other text of any kind.
```

---

**Game Modification Prompt**

```
Your task is to mutate a game written in the Ludii game
description language to produce a new game. Use the following
games as reference for proper Ludii syntax and game structure:

=====Game 1======
{REFERENCE_GAME_CODE}
==========

.
.
.

=====Game i======
{REFERENCE_GAME_CODE}
==========

Now, create a modification of the following game. Make sure to
obey the constraints of the Ludii grammar to create a
syntactically-valid games. In addition, make sure to modify at
least part of the game so that it becomes a new game. Do not
simply copy an existing game.

=====Game to modify=====
{GAME_CODE_TO_MODIFY}
==========

=====Modified game=====
```

---

# F GAVEL Exemplars

**Havabu**

This game is a variant of *Havannah*. Players take turns placing pieces on a square board. If a player manages to form a line of pieces (including diagonals) that connects three different sides of the board, they win. For the purposes of this condition, corners of the board do not count as belonging to any side – however, connecting two corners also results in a win. Finally, a player loses if they do not have any valid moves. Players are also restricted in that they cannot place a piece in any of the 8 squares adjacent to the previous move. It is available to play here: `https://ludii.games/details.php?keyword=Havabu`

```
(game "Havabu"
    (players 2)
    (equipment {
        (board (square 8))
        (piece "Marker" Each)
    })
    (rules
        (play
            (move Add (to (sites Empty)
                if:(not (is In (to) (sites Around (last To))))
                )
            )
        )
        (end
            (if
                (or {
                    (no Moves Next)
                    (is Connected 3 SidesNoCorners)
                    (is Connected 2 Corners)
                })
                (result Mover Win)
            )
        )
    )
)
```

**YavaGo**

This game is a variant of *Yavalath*. Players take turn placing pieces on a hexagonal grid. If a player gets five pieces in a row, they win. However, if a player gets four pieces in a row, they lose (meaning that five-in-a-row lines must be constructed out of two smaller pieces). In addition, completely surrounding one or more of your opponent's pieces causes them to be removed from the board. It is available to play here: `https://ludii.games/details.php?keyword=YavaGo`

```
(game "YavaGo"
    (players 2)
    (equipment {
        (board (rotate 90 (hex 5)))
        (piece "Marker" Each)
    })
    (rules
        (meta (no Repeat))
        (play
            (move Add
                (to (sites Empty))
                (then
                    (enclose
                        (from (last To)) Orthogonal
                        (between if:(is Enemy (who at:(between)))
                            (apply (remove (between)))
                        )
                    )
                )
            )
        )
        (end {
            (if (is Line 5) (result Next Loss) )
            (if (is Line 4) (result Next Win))
        })
    )
)
```

**HopThrough**

This game is a variant of *Breakthrough / Knightthrough*. Players start with two rows of pieces on opposite sides of the board, and the objective is to have a piece reach the other end. Pieces can only move by hopping over one another (either laterally or diagonally), and there are no captures. It is available to play here: `https://ludii.games/details.php?keyword=HopThrough`

```
(game "HopThrough"
    (players 2)

    (equipment {
            (board (square 8))
            (piece "Counter" Each
                (move Hop
                    (between if:(is Occupied (between)))
                    (to if:(is Empty (to)))
                )
            )
            (regions P1 (sites Top))
            (regions P2 (sites Bottom))
    })

    (rules
        (start {
                (place "Counter1" (expand (sites Bottom)))
                (place "Counter2" (expand (sites Top)))
        })
        (play
            (forEach Piece)
        )

        (end
            (if (is In (last To) (sites Mover)) (result Mover Win))
        )
    )
)
```

