# OpenReview forum: "GAVEL: Generating Games via Evolution and Language Models"
_NeurIPS.cc/2024/Conference — NeurIPS 2024 poster_

### Official Review · Reviewer_shZT · 2024-06-23

**Soundness:** 3
**Presentation:** 3
**Contribution:** 1
**Rating:** 5
**Confidence:** 4

**Summary:**

The paper presents a method to generate board games in a domain specific language using an LLM tuned on that language. An LLM is finetuned using fill-in-the-middle training to complete descriptions of board games from a previously gathered dataset. This LLM is used as a mutation operator in a quality-diversity algorithm. Evaluations start generation from a held-out set of seed games and show the algorithm produces more variation of high quality than an ablation. Preliminary expert human evaluations are favorable for the quality of the newly generated games.

**Strengths:**

# originality
Low
- Using LLMs as mutation operators was previously established as a methodology (as referenced).
- Fine-tuning LLMs to produce domain-specific code is also well-explored.
- The primary novelty is the target dataset used.

# quality
Reasonable
- Great to have statistical testing of differences!
- The new domain makes it hard to have alternative baselines, but this makes it difficult to know how successful the algorithm is at solving the base domain task. The quality-diversity metric is "internal" to the algorithm in the sense of being a proxy metric to produce the desired "good games." The preliminary expert evaluations are promising, but not a strong piece of evidence (yet).


# clarity
Good
- Clearly articulates the problem and approach. Introduces domain-specific information to help readers understand the DSL and its features.
- The paper is direct about limitations and potential extensions.


# significance
Modest
- Of interest to the game generation community.
- Potentially of some interest to the code generation community, but the techniques are not particularly novel.
- The expert human evaluation of potentially novel games is a positive step toward impact.

**Weaknesses:**

Fundamentally the paper rests on the success of the algorithm at generating games (given the relatively established techniques being employed). This is hard to gauge from the experiments so far: the QD metric does not directly measure success against a ground truth and the comparison to GAVEL-UCB is an ablation (thus relative change) without baseline algorithms to compare against.

How could the paper provide more persuasive evidence that GAVEL is advancing the problem of board game generation?
Perhaps consider comparing to few-shot prompted code LLMs as a more "naive" baseline. How does GAVEL compare in terms of number of tokens needed to produce games and the quality of the best of those games?
Or perhaps there is a simple ablation that replaces MAP-Elites with a reasonable alternative and shows that GAVEL produces more diverse and/or better games when consuming the same number of steps of generation (or raw tokens produced through LLM inference). These experiments (unfortunately perhaps too much for the discussion period) would make it clearer that the more direct ways to solve this problem are not sufficient. Matching on inference (token) budget would further provide evidence of better efficiency. Or perhaps this would show GAVEL can asymptote to greater quality in final artifacts even if it demands a larger inference budget.

For the human evaluation, is there recorded feedback or other narrative information from responses that might strengthen the claims that GAVEL is succeeding?

My core concern is the techniques used are not particularly novel in isolation and in combination, but the domain is too novel to know if the algorithm is succeeding from the experiments so far. I would be persuaded to increase my score if I saw stronger results that GAVEL is truly solving game generation, or is a superior approach to reasonable alternatives.

**Questions:**

- Section 3.1
	- How many games were filtered by each step of the process? It looks like ~1/2 the games were filtered (from 1182 to 574+14).
- Section 4.2
	- How is the concept vector defined for new games? Is it derived from the seed game concept vector (potentially mutated)?
- (minor) lines 225-227: "In cases like ours where fitness values can be negative, each fitness value is incremented by the minimum possible fitness (i.e. -2) before being summed to ensure that the QD score increases monotonically over time."
	- Perhaps this is a confusion on my part: if the minimum is fixed and samples are only added to the archive when increasing in fitness, this (seems to) imply that the score can only increase. Offsetting by the minimum may shift the minimum value obtained to be positive, but should not change the process of monotonic improvement.

**Limitations:**

Yes. The paper describes the limitations of the current approach and provides potential extensions to mitigate these weaknesses. This includes generating unused code, limited diversity, construction of the archive, and automating human evaluation.

---

> ### Author Rebuttal · Authors · 2024-08-06
>
> We thank the reviewer for their careful comments and critiques. We hope to address their primary concern through additional baseline experiments that were performed during the response period.
>
> Specifically, we compare GAVEL against directly sampling from our fine-tuned language model and few-shot prompting with GPT-4o. For the direct sampling approach, we randomly select a game from the validation set and mutate it using the same selection and re-sampling process and repeat this process 4500 times (this is to mirror of the 500 generations of GAVEL, since in each generation we randomly select 3 games and perform 3 mutations on each). We then evaluate each of the generated games for fitness and determine the cell they would occupy based on their concepts. This allows us to construct a MAP-Elites archive (i.e. by  keeping the most fit game in each cell) for the baseline in order to directly compare against GAVEL. We perform a similar process for GPT-4o, except we provide each of the 14 validation games as part of the prompt and ask the model to create a modification of one randomly-selected validation game. Due to constraints on budget we obtain only 900 samples from GPT-4o and thus compare to GAVEL after 100 generations. In a camera-ready version of the paper we will extend the GPT-4o baseline to the full 4500 samples and experiment with including additional context in the prompt.
>
> The results of these baseline experiments are presented in the accompanying PDF, but the short summary is that both baseline methods fall far short of GAVEL in terms of QD score. Pure sampling appears to obtain dramatically fewer high-fitness samples both across the archive and in new cells. While GPT-4o produces a similar number of high-fitness samples in new cells it appears to produce dramatically fewer high-fitness samples overall. In both cases, it seems that GAVEL is doing a better job exploring the space of possible games. We will include these results in a camera-ready version of the paper.
>
>
> In addition, we provide individual responses to the specific questions raised:
>
> - Q1: How many games were filtered by each step of the process?
>
>     - A: From the 1182 games in the full dataset, we first filter down to 1033 games in the “board” category. Removing the Mancala-style games brings the dataset to 825 games, and filtering by tokenized length brings us to our final 574 training games and 14 validation games.
>
> - Q2: How is the concept vector defined for new games?
>
>     - A: Ludii can automatically compute the concept vector for any game that successfully compiles: each feature is derived from the combination of “ludemes” (essentially, keywords) that are used in the game description.
>
> - Q3: Doesn’t the QD score increase monotonically even without the offsetting of individual fitness values?
>
>     - A: In most MAP-Elites implementations (and in ours), a sample can get added to a previously-empty archive cell regardless of its fitness score. This means that, in our case, games in novel cells with negative fitness scores can cause the QD score to decrease. We will make this distinction more clear in the camera-ready version of the paper.

---

> > ### Comment · Reviewer_shZT · 2024-08-11
> >
> > Thank you for the direct answers and additional experiments!
> >
> > I'm persuaded by these results that naive baselines would not produce as good results in the chosen metrics. I remain somewhat concerned that QD score is not necessarily a good metric for the objective of the work and that MAP-Elites is designed to optimize the score (thus should do better on this rating). That said, the results are clear that random sampling and even a "simple" LLM (GPT4o being SOTA at this time!) show that the choice of both Map-Elites (vs random sampling) and FITM (vs GPT4o) are good choices for this problem.
> >
> > Overall this raises my score.
> >
> > > filtering by tokenized length brings us to our final 574 training games
> >
> > Thank you for the details on the filtering! Hopefully improvements in LLM context lengths will yield improvements to the FITM training and baseline LLM being used. (This is not something I expect for this submission, but a comment about future developments of the work)
> >
> > > Q2
> >
> > Ah, I see. That remains a bit opaque, but I'm comfortable with that given it's a detail others would only need to know if reproducing without rerunning the code. It may help to detail that in the appendix for the sake of reproducibility.
> >
> > > Q3
> >
> > Thank you for the clarification; that makes sense.

---

### Official Review · Reviewer_ocXt · 2024-07-05

**Soundness:** 3
**Presentation:** 3
**Contribution:** 3
**Rating:** 5
**Confidence:** 4

**Summary:**

The paper presents GAVEL (Games via Evolution and Language Models), a system designed for automated game generation. The authors utilize the Ludii game description language (L-GDL) to encode game rules and leverage a combination of evolutionary computation and large language models (LLMs) to generate new and novel games. The system is evaluated both quantitatively and qualitatively, demonstrating the capability to produce playable and interesting games distinct from those in the existing Ludii dataset.

**Strengths:**

- The integration of evolutionary algorithms with large language models for game generation is a promising approach, expanding the capabilities beyond traditional rule-based or heuristic methods.
- The paper provides a thorough evaluation of the generated games, using both automated metrics and expert human analysis, which strengthens the validity of the results.
- The paper clearly describes the training process of the language model, the evolutionary search algorithm, and the evaluation metrics, allowing for reproducibility and further exploration by other researchers.

**Weaknesses:**

-  While innovative, the application of this research might be seen as niche within the broader NeurIPS community. The practical impact and scalability of such a system to other domains should be further tested in the future. Additionally, the impact on the broader ML community should be further highlighted.
- There exists some related work on LLM and created games and level, also including combining LLMs with more open-ended search methods like novelty search, that seems missing, e.g.
        - Sudhakaran, Shyam, et al. "Mariogpt: Open-ended text2level generation through large language models." Advances in Neural Information Processing Systems 36 (2024).
        - Todd, Graham, et al. "Level generation through large language models." Proceedings of the 18th International Conference on the Foundations of Digital Games. 2023.

**Questions:**

- What do the authors imagine could be the impact of the work on the larger ML community
- Which other application areas outside of games could the system be used for?

**Limitations:**

Yes.

---

> ### Author Rebuttal · Authors · 2024-08-06
>
> We thank the reviewer for their careful comments and critiques and we hope to address their general concerns in our response. We also appreciate the pointers to additional related work and will include them in a camera-ready version of our paper.
>
> While it is the case that the results we present are specific to games and the Ludii description language, we feel that our approach is general enough to apply to most domain-specific languages. DSLs are used in a variety of contexts and many of them are not well-represented in the training data of LLMs -- motivating a fine-tuning approach. Our fitness function is also general-purpose enough that it would work with minor modifications for other game-generation domains and the broad approach of a heuristic fitness function should in principle be applicable to an even wider range of domains. In this way, the broad GAVEL approach could be adapted for effective code synthesis in relatively data-sparse settings.
>
> However, even if our method only works effectively for board game generation, this would in our opinion be both significant and general. Most of the world’s population play board games, at least occasionally, and the Ludii dataset contains more than a thousand games from all over the world. In the sense that our model is tasked with producing modifications to an existing game, it has demonstrated a broad and wide-ranging ability to do so. Further, the procedural generation of novel games presents an avenue for future research in open-ended learning.
>
> Finally, we would also like to press the significance of the empirical results of our paper. We are the first to generate not one, but several new and interesting board games. The only directly comparable work is Cameron Browne’s Ludi system from 2009, but that operated in a significantly smaller search space consisting only of connection and line-completion games.

---

> > ### Comment · Reviewer_ocXt · 2024-08-12
> >
> > Thank you for the clarifications. Given the additional experiments and results, I'm happy to increase my score. I still think it would be good to extend the related work sections with other relevant work on LLMs and games. [As a side note, I would suggest not posting to arxiv during the review period since it makes the double blind review process more difficult]

---

### Official Review · Reviewer_HLJN · 2024-07-11

**Soundness:** 3
**Presentation:** 4
**Contribution:** 2
**Rating:** 6
**Confidence:** 4

**Summary:**

The paper targets on generating interesting games automatically. To do this, it proposes an evolution-based algorithm, which iteratively mutates the game components using the generalizability of an LLM. The work is based on a previous large-scale game datasets Ludii.

**Strengths:**

1. It is an interesting task to me, to automate the process of game design using AI. The authors make a good job in presentation. Some of the details really enlighten me.

2. The authors leverage LLMs to evolve the games iteratively. This is a nice idea, or at least a nice application of today's AI.

3. The word is solid. The experiments are based 1000 different games. While the games are limited in board games, it is still a great job and solid work.

**Weaknesses:**

1. **Lack of illustrative cases of generated games.** I am interested in how the generated games vary in the process of evaluation (maybe a case in generation 10 and another case in generation 100?). Second, the cases provided in the paper are in Ludii language. It is hard for readers to read immediately. It is better to provide at least a piece of natural language description.

2. **Evaluation is fair.** The evaluation is ok to me, while not novel enough. Most results are predictable and lack of inspiration.

3. **It is still a hard task to evaluate the degree of interest of a game.** The authors propose a number of approaches to do that. Though these approaches are sound to me, most of them have been discussed in previous work. Recent approaches are still far from a faithful and comprehensive evidence for a game being interesting.

4. **Some training and evaluation details are missing.** Please see below.

Despite some limitations, I still believe that this paper offers a solid and interesting work for specific communities.

**Questions:**

1. Is the used CodeLLaMA model the base-version or instructed version? Or which one performs better in your fine-tuning process?

2. The authors mention that the LLM would perfectly reproduce the missing section of code in the process of mutation. The proposed solution is to mutate a set of unseen games. It is surprised to me that the authors do not try to use top-k decoding or noisy decoding, since these methods are more intuitive and easy to me. Or these methods are not good in practice?

3. I am not sure whether this work or the proposed method is greatly reliant on a high-level descriptive game language, or not? If we put this work in broader scenes, e.g. role-play games, do the principles and algorithms in the paper also work?

4. It is not clear how to judge the correctness of the generated code, since the new games are represented in code (if I am wrong please correct me). Solely checking the compilation is not enough to determine if the code is right or not.


5. I hope the authors can discuss more recent work in AI generating games, e.g. Instruction-Driven Game Engines on Large Language Models (https://arxiv.org/abs/2404.00276). In this work, the authors automate the development process of poker games based on LLMs.

**Limitations:**

Yes.

---

> ### Author Rebuttal · Authors · 2024-08-06
>
> We thank the reviewer for their insightful comments. We acknowledge the reviewer’s points about the difficulty of evaluation and will include a more thorough discussion of these challenges in a camera-ready version.
>
>
> We also provide individual responses to the specific questions raised:
>
> - Q1: Is the used CodeLLaMA model the base-version or instructed version? Or which one performs better in your fine-tuning process?
>
>     - A: We used the base version since we were fine-tuning on a single task. However, in some of our early experiments, we also fine-tuned instruct models to translate board game instructions from natural language to the Ludii DSL. While we eventually dropped that line of research due to the poor quality of the English descriptions accompanying Ludii games. We didn’t notice any significant difference in performance between the fine-tuned instruct and non-instruct versions of CodeLLaMA.
>
> - Q2: The authors mention that the LLM would perfectly reproduce the missing section of code in the process of mutation. The proposed solution is to mutate a set of unseen games. It is surprised to me that the authors do not try to use top-k decoding or noisy decoding, since these methods are more intuitive and easy to me. Or these methods are not good in practice?
>
>     - A: We agree that diversity-boosting strategies like top-k or noisy decoding are intuitive for increasing the novelty of mutations. However, our early experiments showed that these decoding strategies were ineffective at generating mutations that were both novel and valid. Given our relatively small pool of initial seed games, we found that simply excluding them from the training data was more effective. We will clarify this point in the camera-ready revision of the paper.
>
> - Q3: I am not sure whether this work or the proposed method is greatly reliant on a high-level descriptive game language, or not? If we put this work in broader scenes, e.g. role-play games, do the principles and algorithms in the paper also work?
>
>     - A: The approach we propose is applicable to most formal languages, including DSLs and general-purpose programming languages. The main constraint is that candidate games (or programs) need to be evaluated using a reward function. We achieve this by compiling the game descriptions, playing through them, and using heuristics to quantify the quality of the playthrough. Unlike strategy games, defining similar heuristics for games that feature a stronger human element, such as role-playing games, may be more challenging.
>
> - Q4: It is not clear how to judge the correctness of the generated code, since the new games are represented in code (if I am wrong please correct me). Solely checking the compilation is not enough to determine if the code is right or not.
>
>     - A: Indeed, compilability is only one of the metrics we consider when evaluating the quality or correctness of a game. Instead, we focus on evaluating the quality of playthroughs. We compile sample games into executable programs (they compile to Java bytecode), collect a series of playthroughs using self-play between MCTS agents, and evaluate each game’s playthroughs in aggregate using heuristics that quantify the game’s balance, decisiveness, completion, agency, and coverage. You can find more details about this process on pages 5 and 6.
>
> - Q5: I hope the authors can discuss more recent work in AI generating games, e.g. Instruction-Driven Game Engines on Large Language Models (https://arxiv.org/abs/2404.00276). In this work, the authors automate the development process of poker games based on LLMs.
>
>     - A: We thank the reviewer for pointing us towards this paper. While its approach is quite different from ours - they use LLMs as a game engine to predict next states, while we use them to define rulesets in a DSL - the authors' strong results are exciting. We will consider including it as related work in the camera-ready version of the paper.

---

> > ### Comment · Reviewer_HLJN · 2024-08-12
> >
> > Thanks for further explanations. I will keep my rating and support acceptance.

---

### Official Review · Reviewer_29DD · 2024-07-12

**Soundness:** 2
**Presentation:** 4
**Contribution:** 2
**Rating:** 5
**Confidence:** 5

**Summary:**

The authors consider the problem of generating sets of diverse and interesting multi-player games. They instantiate this problem in the subspace of board-game like games using a recent domain-specific language called Ludii. They then use a quality-diversity algorithm (specifically, map-elites with a language-model-based mutation operator) as their generation algorithm. They name their overall approach GAVEL.

For the mutation operator, the authors fine-tune a 13b codellama model with a fill-in-middle objective to reconstruct structured portions of Ludii games in a train dataset of ~500 games. For the behavioral characteristic, the authors use a set of game features provided by the game engine on which they have performed PCA (fitted on the train set). For the fitness function, the authors use a set of 3 filters allowing to remove invalid, trivial, severely unbalanced games (among others) and then compute separate scores along several metrics of playablility derived from domain insight and expertise (e.g. amount of time a player has more than 1 option to play, proportion of games that don't end in a draw, etc). The map-elites archive is seeded with 14 games novel to the mutation model.

The authors report qualitative as well as quantitative evaluations of map-elites on their domain. They demonstrate that the method outperforms (by QD-score) a version where portions of code that have been successfully mutated beforehand are preferentially mutated (GAVEL-UCB). They show that GAVEL is able to create games in novel cells compared with the original Ludii set. They showcase some their created games, selected from the set of promising games by human experts.

**Strengths:**

* The paper tackles automatic game generation, which is an interesting area of research where modern machine learning methods have much to contribute;
* The paper is excellently written and very clear (to a reader with knowledge of the area at least). Presentation, pacing and discussion are well executed; it was a pleasure to read. The illustrations are nice;
* The methodology is very sound, all steps are explained in detail and would be simple to reproduce.
* The application of evolution through large model (ELM) like methods to game generation seem novel;
* The cherry-picked examples of games created by the method look engaging, demonstrating the interest of the method;
* The use of concept vectors are a representative behavioral characteristic (BC) in this domain;
* The fitness function, while domain-specific, is precise and captures fine-grained notions of enjoyableness of games;
* Several seeds of all experiments are given, allowing to quantify variability;
* The authors are very upfront about the limitations of their methods and the claims being made are well supported with evidence.

**Weaknesses:**

* I worry that the contribution of the paper is limited. While using Map-Elites + LLMs on Ludii is novel, both the fitness function and the BC space used are very specific to games (of the kind presented in the paper), and the paper presents no algorithmic improvement. I would have welcomed methods that work across domains (for instance ask an LLM to self-refine a fitness function and BC space?)
* While generally, applying a method to a novel domain is interesting in its own right, I worry about the lack of comparative studies in the experiments of the paper. It is good to study map-elites on this new domain but what are the lessons learned? Where are comparisons to other methods, could other generation algorithms (like the ones cited in the related work, or sampling from the model repeatedly without QD, or few-shot prompting a large model with a larger context window) function just as well? Right now this paper reads more like a technical report than a scientific paper, since I do not feel I learned how this complex algorithm works on this domain by careful ablations and comparisons.
* There could be more information on the study of dynamics of evolution of games in the paper. Why did the authors stop at 500 evolution steps? Was the discovery process saturating? If yes, why? Could more OOD games have been discovered if the process would have been run for longer? If more samples would have been used to seed the archive?

**Questions:**

### Questions

* line 104 did you try not removing these games, does anything bad happen?
* would CVT-mapelites have worked in this case (to avoid training the PCA on train set games and collapsing variation that doesn't exist in the train set?)
* Do you have ideas on how to characterize game diversity in other spaces than the BC space? How can we validate that this space adequately capture what humans mean by diverse games?
* finetuning vs in-context learning: why have you fine-tuned a smaller model instead of asking a large, capable instruction model to perform the task (with examples?) Did you experiment with this? (cost is a valid reason to try using a smaller model, but I am curious on how larger models perform on the task).
* Open question on fitness:  could the various playability measures be collapsed to a single metric such as learnability, or a computational measure of interestingness, do you have ideas on how this could work or whether it is feasible? (See for instance the interesting but empirically ungrounded work in https://arxiv.org/abs/0812.4360, somewhat related to a recent formulation of open-endeness: https://arxiv.org/abs/2406.04268). This is beyond the current scope of the paper, but I am curious to know if the authors have any thoughts as to how this relates to game domains.
* Why to you think the UCB variant works less well?
* The limitation section discusses biasing mutation, do you have any idea on how this could work in practice?

### Suggestions

* (minor) The middle figure of figure 2 could show function expansion;
* l121 should have a footnote specifying when the training was performed because these things change very fast. (eg codestral can do fill-in-the-middle too now, but was released after the neurips deadline);

### Related work

Here is some work related to the current approach:
* There is a recent trend for game generation with LLMs, mostly focused on generating geometrical information. The seminal work here is MarioGPT (https://arxiv.org/abs/2302.05981) but there have been several works in this vein since, like https://arxiv.org/abs/2403.12014, https://arxiv.org/abs/2302.05817 or https://arxiv.org/abs/2406.04663
* On LLM-augmented QD, rainbow-teaming is also an important paper https://arxiv.org/abs/2402.16822

**Limitations:**

The authors acknowledge both a large set of limitations of their work as well as its broader impact. I would add that an important limitation of the paper is that the proposed implementation of LLM-augmented map-elites lacks generality: fitness, mutation operator and BC all are specific to the Ludii environment.

---

> ### Author Rebuttal · Authors · 2024-08-06
>
> We thank the reviewer for taking the time to carefully appraise our work. We hope we can adequately address their comments  here.
>
> First, while it is the case that our results are specific to games and the Ludii description language, we feel that our approach is general enough to apply to most domain-specific languages. DSLs are used in a variety of contexts and many of them are not well-represented in the training data of LLMs -- motivating a fine-tuning approach. Our fitness function is also general-purpose enough that it would work with minor modifications for other game-generation domains and the broad approach of a heuristic fitness function should in principle be applicable to an even wider range of domains.
>
> Next, we take the reviewer’s point about the lack of comparative studies and attempt to address it through additional baseline experiments. Specifically, we compare GAVEL against directly sampling from our fine-tuned language model and few-shot prompting with GPT-4o. For the direct sampling approach, we randomly select a game from the validation set and mutate it using the same selection and re-sampling process and repeat this process 4500 times (this is to mirror of the 500 generations of GAVEL, since in each generation we randomly select 3 games and perform 3 mutations on each). We then evaluate each of the generated games for fitness and determine the cell they would occupy based on their concepts. This allows us to construct a MAP-Elites archive (i.e. keeping the most fit game in each cell) in order to directly compare against GAVEL. We perform a similar process for GPT-4o, except we provide each of the 14 validation games as part of the prompt and ask the model to create a modification of one randomly-selected validation game. Due to constraints on budget we obtain only 900 samples from GPT-4o and thus compare to GAVEL after 100 generations.
>
> The results of these baseline experiments are presented in the accompanying PDF, but the short summary is that both baseline methods fall far short of GAVEL in terms of QD score. Pure sampling appears to obtain dramatically fewer high-fitness samples both across the archive and in new cells. While GPT-4o produces a similar number of high-fitness samples in new cells it appears to produce dramatically fewer overall. In both cases, it seems that GAVEL is doing a better job exploring the space of possible games. We will include these results in a camera-ready version of the paper.
>
> Finally, we provide individual responses to the specific questions raised by the reviewer:
>
> - Q1: Did you try not removing the Mancala-style games?
>
>     - A: In initial experiments, our fitness evaluation had difficulty getting reasonable approximations for these Mancala-style games in the limited computational budget we provided. In addition, these games represent a fairly distinct cluster in terms of concepts of ludemes used and we omitted them in order to increase the relative “generality” of the training dataset, though future work could include them.
>
> - Q2: Would CVT-MAP-Elites have worked in this case?
>
>     - A: This is a very reasonable idea, and we did explore CVT-MAP-Elites in an initial experiment as well. Unfortunately running the CVT algorithm over the full concept vector space produced either a very sparse archive (when the number of cells was high) or a relatively collapsed archive (when the number of cells was small), which we attribute to the fact that the tessellation occurs uniformly over the space while the games are not distributed uniformly.
>
> - Q3: Do you have ideas on how to characterize game diversity in other spaces than the BC space?
>
>     - A: Capturing human notions of diversity in gameplay remains an open research question. Prior work does indicate that Ludii concepts capture certain high-level judgements of game genre (see citation [54] in our paper), but outside of this the gold standard would likely remain a human user study. We are in the process of organizing such a user study for future work / a camera-ready version of this paper.
>
> - Q4: Why have you fine-tuned a smaller model instead of asking a large, capable instruction model to perform the task?
>
>     - A: See baseline experiment above
>
> - Q5: Could the various playability measures be collapsed to a single metric such as learnability, or a computational measure of interestingness?
>
>     - A: While this idea is reasonable, we think that the main challenge would be in tractability. Using a measure of learnability as a fitness function would effectively add an inner loop to what is already the main bottleneck of the evolutionary algorithm. Perhaps such an approach would work in domains where generated game code could be compiled to something that runs on the GPU, but it would likely be infeasible for Ludii.
>
> - Q6: Why do you think the UCB variant works less well?
>
>     - A: One hypothesis is that it is very difficult to make any kind of improvement in the fitness landscape GAVEL operates over. The UCB MAP-Elites selector relies on the intuition that cells which produced successful mutations in the past are more likely to do so again. However, if improvement in any given cell is basically equally (un)likely, then the UCB algorithm would effectively waste time re-selecting previously successful cells instead of a more efficient uniform sampling.
>
> - Q7: The limitation section discusses biasing mutation, do you have any idea on how this could work in practice?
>
>     - A: In the context of Ludii, the most likely way to accomplish this would be with a heuristic-based method. It should in principle be possible to write a set of rules that link certain kinds of errors (e.g. unused components in the “equipment” section) with certain program regions (e.g. player actions in the “rules” section). Another possibility is an instruction-based mutation operator to which additional context (e.g. “this game has unused components”) could be provided.

---

> > ### Comment · Reviewer_29DD · 2024-08-08
> >
> > I thank you for your response, it answers all of my questions and clarifies some points. I am very happy that you performed additional experiments with gpt4 and repeated sampling from the model and I think this strengthens the paper.
> >
> > I still feel that my point on the generality of the method stands (as well as the fact that LLM-augmented map-elites is not new), so I will not be raising my grade.

---

### Author Rebuttal · Authors · 2024-08-06

We would like to thank each of the reviewers for taking the time to consider our work. In the attached PDF we include the results of additional baseline experiments performed during the response period that compare GAVEL to a pure sampling approach and few-shot prompting with GPT-4o. The details and take-aways of these experiments are included in the table captions.

---

### Decision · Program_Chairs · 2024-09-25

**Decision:**

Accept (poster)

**Comment:**

An innovative paper, rated highly for clarity, on the very timely topic of generating many novel and diverse games using code-LLMs. The reviewers are overall positive, and I too see a large potential for this work, both because generating interesting games is valuable in itself, and because I can see the method's components transfer to related domains of generating tasks or code. I'm looking forward to the major revisions in camera-ready with the additional baselines, the user-study, and an appendix that documents (or revisits) all these mentioned "initial experiments" mentioned in the author feedback.